# Impact of Dietary-Forage-to-Concentrate Ratio on Podolian Young Bulls’ Performance and Nutritional Properties of Meat

**DOI:** 10.3390/ani15020166

**Published:** 2025-01-10

**Authors:** Rosaria Marino, Mariangela Caroprese, Antonella Santillo, Agostino Sevi, Marzia Albenzio

**Affiliations:** Department of Agriculture, Food, Natural Resources and Engineering (DAFNE), University of Foggia, Via Napoli, 25-71121 Foggia, Italy; rosaria.marino@unifg.it (R.M.); antonella.santillo@unifg.it (A.S.); agostino.sevi@unifg.it (A.S.); marzia.albenzio@unifg.it (M.A.)

**Keywords:** carnosine, creatine, fatty acids profile, forage-to-concentrate ratio, oxidative stability Podolian meat

## Abstract

The nutritional properties of beef are very notable for purchasing behaviour, because consumers are interested in health warnings. Improving the nutritional properties of meat through animal feeding strategies is crucial in order to meet current human nutritional recommendations. This study demonstrates that a high forage-to-concentrate ratio for finishing Podolian young bulls enriches the meat with healthy fatty acids, bioactive compounds, and greater oxidative stability, which is fundamental for a meat that needs more than two weeks of aging. This may help to promote the sustainability of Podolian beef production systems contributing to the conservation of semi-natural habitats, as well as to healthy meat production.

## 1. Introduction

In ruminant rearing systems, improving the efficiency of animal production is one of the main objectives of producers; dietary composition and nutrient availability can affect energy utilisation and partition [1]. Traditionally, the ruminant diet comes from grazing pasture, as their digestive systems have evolved to use forage. Many studies have shown that forage finishing results in leaner carcasses compared with grain finishing [2,3]. In a previous study, Marino et al. [4] found that increasing the concentrate’s proportion to 40% in rations administered to Podolian young bulls, which had access to paddock during the finishing period, led to an increase in the growth rate, but it did not change the carcass yield and quality. However, it is not easy to define what constitutes a low- or high-concentrate diet, as it differs depending on many factors such as the production system and breed reared.

Podolian cattle are an important breed reared in Italy, which, after a reduction in their numbers over the last years, is now showing an increase (37,312 Podolian cattle [5]). It is a rustic breed that is widespread in Southern Italy, well adapted to the difficulty of the surrounding environment, and is characterised as producing meat with beneficial contents of polyunsaturated fatty acids (PUFAs) [6]. Although Podolian cattle exploit pasture, the use of concentrate supplementation is necessary, especially during some periods of year, to increase cattle’s body weight, because of the poor pasture quality in most areas of Southern Italy (<6% CP, DM basis) due to low summer rainfall. Therefore, it could be useful to boost the genetic growth potential of the Podolian breed through the administration of a relatively high concentrate level of diet supplementation during the finishing period, which usually lasts 4–5 months, because of the rusticity and growth metabolism of this breed. The forage-to-concentrate ratio is very important in cattle rearing systems for beef production, affecting diet digestibility [7], animal health [8], and the health characteristics of meat [9]. Reconciling the feeding of high amounts of concentrate to support a high production performance with animal health and meat quality represents a challenge in the current ruminant feeding system. Meat is a superb nutritional component that provides protein with high biological value, vitamins, minerals, and bioactive compounds that contribute to human health [10]. Particularly, creatine exhibits important physiological activity, playing key roles in muscle energy metabolism and neurodegenerative diseases such as Parkinson’s and Huntington’s [11], whereas carnosine and anserine exert antioxidants and antiaging effects [12]. Furthermore, in vitro and in vivo studies via animal models or human clinical trials report that many of the meat-derived biomolecules are multifunctional and can engage in more than one of the health-related activities mentioned [13].

Many studies have focused on the effect of the forage-to-concentrate ratio on the fatty acid profile of meat, highlighting that high-forage diets for finishing beef cattle increase the deposition of biohydrogenation intermediates and n-3 poly-unsaturated FAs [5,14]. However, little is known on how various forage-to-concentrate ratios affect the contents of other components of beef such as bioactive compounds, including carnitine, carnosine, and anserine. In addition, Podolian meat is often characterised by reduced tenderness because of the low presence of marbling and increased physical exercise of animals because of the extensive rearing system; in such conditions, extended post mortem aging is needed to improve tenderness of this type of meat [15]. Therefore, the aim of this study was to assess the impact of various dietary forage-to-concentrate ratios (65:35 vs. 45:55) on the fatty acid profile and bioactive compounds of Podolian meat aged for two different durations (11 vs. 18 days). The levels of the two forage-to-concentrate ratios were chosen on the basis of a previous study [4] and with the aim to improve the sustainability of beef production systems by replacement of protein concentrates, which are expensive, with locally produced forages such as alfalfa hay.

## 2. Materials and Methods

### 2.1. Experimental Design and Animal Management

All animal procedures were approved by the Foggia University Institutional Animal Care and Use Committee (protocol number: 03-2023). The field trial, which lasted 142 days, was conducted from November 2023 to March 2024 on an animal farm located in the Gargano National Park, which is 60 km northwest of Foggia (Southern Italy), with an elevation of about 300 m above sea level.

A total of 20 Podolian young bulls that averaged 413 ± 9.55 (SE) days in age, with mean initial body weights of 326 ± 19.43 (SE) kg, were kept in a barn with free access to a paddock. They were randomly assigned to two groups of 10 each according to the forage-to-concentrate ratio. The high forage-to-concentrate (HF:C) group received a diet with a forage-to-concentrate ratio of 65:35, and the low forage-to-concentrate (LF:C) group received a diet with a forage-to-concentrate ratio of 45:55. Water was always available for all the animals. Before the experimental period, throughout the growth phase, and until the finishing period, the animals grazed on natural pasture. During the experimental period, the animals were fed in individual, covered pens equipped with automatic waterers and individual feeders. All animals were fed twice per day at 8 a.m. and 3 p.m., and the forage and concentrate were simultaneously administrated in cribs and mangers. The feeder space per animal was about 0.6 m.

The diets were balanced according to the protein content to ensure isonitrogenous treatments, and they were calculated to satisfy the growth requirements suggested by the I.N.R.A. [16]. Details on the ingredient sources, chemical compositions, and additives of the diets are reported in Table 1.

Analyses of the feed were carried out on collected pooled feed samples. The neutral detergent fibre and ADF were determined according to Van Soest et al. [17], while crude fat and crude protein were analysed following the Association of Official Analytical Chemists methods [18]. Lipids were extracted using the procedure described by Sukhija and Palmquist [19], and fatty acid methyl esters were identified using the same procedure described for the fatty acids of the meat samples.

### 2.2. Performance and Post-Slaughter Measurements

All animals were individually weighed at fortnightly intervals in order to estimate the average daily gain (ADG, g/day) and the feed efficiency (FE, meat forage units/kg gain). Individual dry matter (DM) intake was calculated daily as the difference between the amount of feed offered and refused. At the end of the finishing period (142 days), all animals were weighed on the morning of slaughter to record the final body weight (BW) and transported to a commercial abattoir. Slaughter occurred according to industrial routines used in Italy and to EU rule n.1099/2009. After slaughter the hot carcass weight was recorded and the dressing percentage calculated.

The *longissimus thoracis* (LT) muscle was removed from the right carcass half, divided longitudinally into 2 sections, and aged in vacuum packaging at 4 °C until 11 and 18 d post mortem. Cranial and caudal sections were randomised across aging treatments. The meat’s chemical composition, fatty acids profile, bioactive compounds, a-tocopherol and oxidative stability were estimated for each aging time.

### 2.3. Meat Chemical Composition and Fatty Acid Methyl Esters Profile

Meat samples were ground to an homogeneous consistency using a food processor. The moisture, protein, lipid, and ash contents were determined according to AOAC methods [18].

Determination of meat fatty acids was performed according to O’Fallon et al. [20] with some modifications, as previously described by Marino et al. [21]. Fatty acids methyl esters (FAMEs) were transferred into a gas-chromatographic vial, and the fatty acid profile was quantified through an Agilent 6890 N instrument (Agilent Technologies, Santa Clara, CA, USA) equipped with an HP-88 fused-silica capillary column (length: 100 m; internal diameter: 0.25 mm; film thickness: 0.25 μm). The operating conditions were as follows: carrier gas (helium) at a constant flow of 1 mL min^−1^; split–splitless injector at 260 °C; split ratio of 1:25; injected sample volume of 1 μL; and FID detector at 260 °C. The temperature program of the column was set to 5 min at 100 °C and then increased to 240 °C (3.5 °C min^−1^) and held for 15 min. The retention time and area of each peak were computed using the 6890 N NETWORK GC system software Rev.b.04.03.

Fatty acids were identified by comparing their retention times with the fatty acid methyl standards (FIM-FAME-7-Mix, Matreya LLC, Pleasant Gap, PA, USA) added to C18:1-11t; C18:2-9c 11t; C18:2-9c 11c; C18:2-9t 11t; and C18:2-10t 12c (Matreya LLC, Pleasant Gap, PA, USA).

Qualitative (PUFAs/SFAs, n6/n3, linoleic/linolenic, %EPA + DHA, unsaturation index) and nutritional (atherogenic and thrombogenic indices, hypocholesterolemic/hypercholesterolemic ratio) indices were calculated according to Marino et al. [22]. Δ5-plusΔ6-desaturase was calculated according to the following equation [23]: (C20:2n-6 + C20:4n-6+ EPA + C22:5n-3 + DHA)/(LA + ALA + C20:2n-6 + C20:4n-6 + EPA + C22:5n-3 + DHA) × 100

### 2.4. Creatine, Creatinine, Carnosine, and Anserine Determination

Creatine, creatinine, carnosine, and anserine were extracted from raw meat and analysed by HPLC according to Mateescu et al. [24]. The chromatographic separation was performed using an Infinity 1260 HPLC (Agilent Technologies, Palo Alto, CA, USA) equipped with a Kinetex HILIC Silica column (Phenomenex, Torrance, CA, USA; 4.6 × 75 mm, 2.6 μm) and a DAD detector (Agilent Technologies). Mobile phases consisted of solvent A, containing 0.65 mM ammonium acetate, at pH 5.5, in water/acetonitrile (25:75, *v*/*v*, and solvent B, containing 4.55 mM ammonium acetate, at pH 5.5, in water/acetonitrile (70:30, *vol*/*vol*). The separation conditions were a linear gradient from 0% to 100% of solvent B for 13 min at a flow rate of 1.4 mL/min. Before each injection, the column was equilibrated for 10 min under the initial conditions. Creatine, carnosine, and anserine were identified at a wavelength of 214 nm, whereas creatinine was at 236 nm. For the quantification, sample peak areas were correlated to standard curves, previously obtained under the same chromatographic conditions.

### 2.5. Determination of a-Tocopherol from Muscular Tissue and of Oxidative Stability

a-Tocopherol was extracted from muscle samples following the method of Piironen et al. [25], with modifications according to Marino et al. [5]. Lipid oxidation was determined by measuring the reaction of malondialdehyde (MDA) with 2-thiobarbituric acid (TBA) and expressed as mg MDA/kg meat [26]. The c index was calculated according to the following equation [27]:(% monoenoic × 0.025) + (% dienoic × 1) + (% trienoic × 2) + (% tetraenoic × 4) + (% pentaenoic × 6) + (% hexaenoic × 8)

### 2.6. Statistical Analysis

All data were subjected to an analysis of variance using the GLM procedure of SAS statistical software [28] (Version 9.4; SAS Institute 2013). Individual young bulls were the experimental units. The model included the fixed-effects of the diet treatment and random residual error. The mathematical model for meat chemical composition, fatty acids profile, bioactive compounds, a-tocopherol, and oxidative stability analysis included fixed effects due to diet, aging, and diet × aging, as well as random residual error. All effects were tested for statistical significance (to *p* < 0.05), and significant effects are reported in all tables. When significant differences were found (at *p* < 0.05, unless otherwise noted), Tukey’s test was performed for multiple comparisons among means.

## 3. Results and Discussion

### 3.1. Animal Performance and Beef Production

The effects of the forage-to-concentrate ratio, in vita and post mortem, on growth performance and on beef production of Podolian young bulls are shown in Table 2.

The different forage-to-concentrate ratio did not influence ADG. This result could be attributed to the good feed efficiency exhibited by Podolian cattle. Italian rustic breeds, including Podolian cattle, are well known to exhibit compensatory growth during the fattening period. To confirm this, Serra et al. [29] found that the Maremmana breed, an Italian rustic breed, performed better than Limousine when grazing in marginal areas. In addition, in the present study, the high level of concentrate in the diet did not stimulate greater growth performances in the Podolian young bulls. As reported by Brown et al. [30], there is considerable diversity among animals in their ability to adapt more readily to high concentrate diets.

Slaughtering, carcass weights and dressing percentage were also unaffected by the forage-to-concentrate ratio. This may be due to the similar dry matter intake and feed efficiency observed in animals from the LF:C and HF:C groups throughout the experimental trial.

It is worth noting that the dressing percentage observed in this study was higher than those found in a previous study on Podolian cattle [5]. This result could be due to the greater slaughter weight reached in the present experiment; indeed, a positive correlation between slaughter weight and carcass yield exists on the basis of a positive allometry coefficient found for these parameters [31]. Previous studies [32,33] observed differences in slaughtering parameters at lower forage-to-concentrate ratios (from 20:80 to 35:60) compared to those tested in this study.

The chemical composition of Podolian meat was, on average, 75.59 ± 0.51% moisture, 1.66 ± 0.11% fat, 22.62 ± 0.18% protein, and 1.08 ± 0.1% ash. The absence of differences among the experimental group was ascribed to the comparable growth performance and muscle development observed throughout the experimental period.

### 3.2. Fatty Acids (FAs) Composition of Longissimus Thoracis Muscle

The FAs profile of LT muscle from Podolian supplemented with different forage-to-concentrate ratios at 11 and 18 days of aging is reported in Table 3.

The diet with the high forage-to-concentrate ratio significantly affected the FAs profile of the meat, which reduced the percentage of SFAs (*p* < 0.001). In particular, meat from animals fed the high forage-to-concentrate ratio showed lower percentages of myristic (C14:0; *p* < 0.01) and palmitic (C16:0; *p* < 0.05) acids compared to the low-forage group. The reduction in SFA content in meat from the HF:C group might be related to a reduction in endogenous FAs synthesis from the rumen of young bulls due to the composition of the diet with high forage, which is richer in unsaturated lipids, particularly linolenic acid, and has less saturated fatty acids compared to LF:C diet. Among SFAs, miristic and palmitic acids are considered to be more detrimental to serum cholesterol levels [34]. Thus, meat from the young bulls finished with the high forage-to-concentrate ratio tended to have a more favourable SFA composition, with a potentially lower impact on human serum cholesterol levels than meat from the LF:C group. How beef from animals fed diets with different forage-to-concentrate ratios impacts serum cholesterol levels in hypercholesterolemic patients needs to be investigated. In vitro and in vivo studies via animal models or human clinical trials could elucidate the mechanisms between improvements in meat nutritional quality and human health.

The high forage-to-concentrate ratio also resulted in a significant increase in the level of intramuscular fat unsaturation. Meat from the HF:C group had higher percentages of monounsaturated (*p* < 0.05) and polyunsaturated fatty acids (*p* < 0.001) compared to the meat from animals fed a diet with a low forage-to-concentrate ratio. Particularly, higher levels of vaccenic acid (C18:1t11; *p* < 0.01) were found in meat from the HF:C group compared to the LF:C group, while no differences in C18:1t10 were detected. A higher percentage of CLA 9c, t11 (rumenic acid-RA; *p* < 0.01) was also found in meat from the HF:C group.

The C18:1 t11 FA is an intermediate product of the biohydrogenation of linoleic (LA, C18:2n-6) and linolenic (ALA, C18:3n-3) acids. Microbial biohydrogenation of LA and ALA by anaerobic rumen bacteria is highly dependent on the rumen’s pH [35], and the rumen’s microbial metabolism is crucial in determining the ideal fatty acid composition in muscle of ruminant products.

In the present study, increases in biohydrogenation intermediates (vaccenic acid and rumenic acid) were found in the meat of the animals fed with higher forage-to-concentrate ratios. Less fermentable sugar and more soluble fibre contents in alfalfa hay likely created an environment with a favourable pH that promoted greater activity by *Butyrivibrio fibrisolovens*, the bacterium involved in ruminal biohydrogenation of linoleic and linolenic acids to vaccenic acid. The differences in the starch and fibre contents of the diets can affect changes in rumen fermentation, small intestinal digestion, and availability of net energy in young bulls [7]. Particularly, our findings indicate that a diet with 65% forage is effective for increasing vaccenic and rumenic acids in Podolian meat.

The increased concentration of vaccenic acid in the meat from the HF:C group, together with the high level of rumenic acid, obtained via stearoyl Co-A desaturase from its precursor vaccenic acid, resulted in high levels of total conjugated linoleic acid (CLA; *p* < 0.05). Over the last years, major attention has been focused on the CLA FA because of its proven biological activity in the prevention of obesity, cancer, diabetes, atherosclerosis, and osteoporosis [36]; the results prove that a diet composed of 65% forage may be a feasible strategy to enrich meat with these FAs.

The diets with a 65 to 35 forage-to-concentrate ratio resulted in significantly higher levels of the PUFA n-3 (*p* < 0.001) within the lipid fraction of the meat, while the n-6 levels remained unchanged. Regarding polyunsaturated n-3 fatty acids, the meat from the HF:C group showed higher percentages of linolenic (C18:3n-3; *p* < 0.01), eicosapentaenoic (EPA-C20:5n-3; *p* < 0.001), docosapentaenoic (DPA-C22:5n-3; *p* < 0.01), and docosahexaenoic (DHA-C22:6n-3; *p* < 0.05) acids compared to the LF:C group. In agreement with a previous study [37], the diet with high levels of forage increased the availability of linolenic acid in muscle, resulting in enhanced elongation and desaturation to synthesise EPA, DPA, and DHA. These n-3 long-chain FAs have different beneficial effects on human health due to their cardiovascular and neuroprotective properties [38]. In order to estimate the ability of Podolian young bulls to synthesise long-chain fatty acids from precursors, the activity of Δ5-plusΔ6-desaturase was calculated. The estimated Δ5-plusΔ6-desaturase activity was significantly higher (*p* < 0.05) in meat from the HF:C group, highlighting that the higher intake of forage affected the mechanisms of desaturase and elongase responsible for the synthesis of n-3 long-chain FAs. It has been shown that competition between n6 and n3 fatty acids exists, and n-3 fatty acids are the preferred substrate in the desaturation and elongation pathway [39].

Based on the FA profile, the qualitative and nutritional indices were calculated (Table 4).

Meat from the animals fed with the higher forage-to-concentrate ratio showed better qualitative and nutritional indices, with a higher P/S (*p* < 0.05), EPA + DHA percentage (*p* < 0.01), unsaturation index (*p* < 0.05), and hypocholesterolemic/hypercholesterolemic ratio (*p* < 0.05) and lower n6/n3 (*p* < 0.01) and atherogenic and thrombogenic indices (*p* < 0.05) than meat from the LF:C group. The diet with a 65 to 35 forage-to-concentrate ratio strongly reduced the n-6/n-3 ratio, reaching values lower than the recommended threshold of 4 [40]. Furthermore, meat from the HF:C group was also characterised by low atherogenic and thrombogenic indices. The atherogenic and thrombogenic indices express the levels and interactions through which some fatty acids can promote or prevent cardiovascular diseases. Particularly, meat from the HF group had values very close to the thresholds suggested as desirable to minimise the risks of cardiovascular diseases for the atherogenic index (AI) at <0.5 and thrombogenic index (TI) at <1.0. Focusing on the relationships between dietary fatty acids and plasma low-density lipoproteins related to hypocholesterolemic fatty acids, the higher values found in the meat from the HF:C group are considered more beneficial for human health, highlighting the different effects of meat on human cholesterol metabolism.

The different aging times (11 vs. 18) did not affect the fatty acid profiles. This could be due to the aging method used in the present study. The Podolian meat was aged in vacuum packaging, so the low oxygen transmission rate of the vacuum packaging bag might have had a positive effect by preserving the fatty acids from oxidative processes, in agreement with Johnson et al. [41].

### 3.3. Creatine, Creatinine, Carnosine, and Anserine Contents of Longissimus Thoracis Muscle

The contents of the bioactive compounds, such as creatine, creatinine, carnosine, and anserine, of the LT muscle from Podolian cattle supplemented with various forage-to-concentrate ratios with 11 and 18 days of aging are reported in Table 5.

In meat from the HF:C group, higher contents of creatine (*p* < 0.001), carnosine (*p* < 0.01), and anserine (*p* < 0.05) were found. The supplemental alfalfa used in this study as forage is rich in essential amino acids, such as histidine, methionine, isoleucine, valine, leucine, and threonine, while the concentrate is lacking. Particularly, histidine is one of two main components of carnosine (β-alanyl-L-histidine) and anserine (β-alanyl-N-methyl histidine), and methionine is essential for creatine synthesis, which is limited when methionine is not provided in the diet. Therefore, the greater contents of these bioactive compounds found in the meat from the HF:C group could be linked to the amino acid composition of the forage. In addition, we suggest that the highest content of n-3 PUFAs present in the HF:C have contributed to the greater contents of creatine, carnosine, and anserine. In a study on growing steers, it was found that dietary supplementation with n-3 fatty acids increased sensitivity to insulin, which led to an increase in protein metabolism and a greater availability of essential amino acids [42].

### 3.4. α Tocopherol Content and Oxidative Stability of Longissimus Thoracis Muscle

The different forage-to-concentrate ratios did not affect the α-tocopherol content of the meat (1.75 and 1.55 ± 0.05 mg/g in the HF:C and LF:C groups, respectively. It is worth noting that, regardless of the forage-to-concentrate ratio, the levels of a-tocopherol recorded in this study were higher than those reported in previous studies at values of 1.36 mg/g in Podolian meat [21] and 1.2 mg/g in Marchigiana meat [43].

The oxidative stability of *Longissimus thoracis* muscle from Podolian cattle supplemented with different forage-to-concentrate ratios at 11 and 18 days of aging is reported in Table 6.

The reaction of MDA with TBARS is widely used for measuring the extent of the oxidative deterioration of lipids in muscle-based foods. In the present study, the MDA values were not affected by the different forage-to-concentrate ratios, which were much lower than the 2 mg MDA/Kg of meat considered as the rancidity perception threshold for consumers [44]. The period of grazing before the finishing phase probably enhanced the levels of antioxidant compounds in both groups, with a subsequent reduction in beef oxidation that began immediately after the start of retail display. It is worth noting that aging also did not affect the MDA values, thus suggesting that both wet aging combined with free access to outdoor paddock led to good oxidability until 18 days post mortem.

As expected the peroxidability index was higher for the meat from the HF:C group, as this meat was richer in long-chain PUFAs. However, despite the higher peroxidability index, meat from the HF:C group exhibited an MDA value comparable to the meat from the LF:C group. This result can be attributed to the high content of a-tocopherol in the meat from both groups, which exerted antioxidant properties, particularly in the HF:C group, counterbalancing the pro-oxidant conditions created by the higher levels of n-3. This result is very remarkable for Podolian meat, which, being characterised by low meat tenderness, requires more than two weeks of aging to achieve tenderness.

## 4. Conclusions

A high forage-to-concentrate diet for finishing Podolian young bulls positively impacts the nutritional properties of the meat by increasing the deposition of healthy bioactive compounds, such as CLA, long-chain n-3 polyunsaturated FAs, carnitine, carnosine, and anserine.

Our results suggest that a diet composed of 65% DM forage improves the sustainability of Podolian beef production systems, producing healthy meat with great oxidative stability, which is fundamental for a meat product that requires an aging time longer than two weeks. This may help to preserve and promote this native breed, contributing to the proper management and conservation of semi-natural habitats, as well as the production of healthy meat.

## Figures and Tables

**Table 1 animals-15-00166-t001:** Ingredients and chemical and fatty acid compositions of the diets.

	Diet	
	HF:C	LF:C
Ingredients, %		
Alfalfa hay	65	45
Concentrate ^1^	30	50
Soybean meal	3.8	3.8
Sodium bicarbonate	0.5	0.5
Vitamin and mineral mix ^2^	0.7	0.7
Chemical composition (% DM)		
CP	12.65	12.58
Ether extract	3.15	3.38
NDF	43.15	32.05
ADF	20.11	14.15
Starch	20.94	37.85
ME (MJ/kg DM)	10.05	12.11
Fatty acids composition (% of total fatty acids)		
C14:0	0.35	0.25
C16:0	15.25	18.75
C18:0	4.89	4.55
C18:1	21.75	18.88
C18:2n-6	35.66	47.89
C18:3n-3	22.54	11.55

^1^ Ingredients: barley, 60%; wheat bran, 30%; maize meal, 5%; and wheat meal and molasses, 5%. ^2^ Ingredients: vit. A, 1,150,000 IU; vit. D, 390,000 IU; vit. E, 50 mg; vit. PP, 8500 mg; vit. B1, 112 mg; vit. B2, 112 mg; vit. B6, 80 mg; vit. B12, 1 mg; d-pantothenic acid, 2400 mg; choline 15,000 mg; iron 150 mg; manganese 800 mg; zinc 2200 mg; cobalt 8 mg; iodine 30 mg; selenium 5 mg; and molybdenum 10 mg.

**Table 2 animals-15-00166-t002:** In vita and post mortem animal performance of Podolian young bulls as affected by the forage-to-concentrate ratio (HF:C = high forage-to-concentrate ratio (65:35); LF:C = low forage-to-concentrate ratio (45:55); mean ± SE).

	Forage:Concentrate			
	HF:C (*n* = 10)	LF:C (*n* = 10)	SEM	Effect, *p*
In vita				
Initial live weight (kg)	327	325	19.43	0.841
Final live weight (kg)	470	476	21.23	0.758
A.D.G. (kg/d)	1.01	1.06	0.04	0.655
D.M.I. (kg dm)	7.25	7.78	0.22	0.711
Feed efficiency (MFU/kg gain)	4.78	4.55	0.15	0.612
Post mortem				
Carcass weight (kg)	261	266	11.93	0.345
Dressing percentage (%)	55.53	55.88	0.47	0.212

A.D.G. = average daily gain; D.M.I. = dry matter intake.

**Table 3 animals-15-00166-t003:** Effect of the forage-to-concentrate ratio (HF:C = high forage-to-concentrate ratio (65:35); LF:C = low forage-to-concentrate ratio (45:55)) and aging time (11 vs. 18 d) on the fatty acid profile (%) of *Longissimus thoraci* muscle from Podolian young bulls (mean ± SEM).

	Forage:Concentrate						
	HF:C (*n* = 10)		LF:C (*n* = 10)			*p*, Effects	
Aging Time (d)	11	18	11	18	SEM	Diet	Aging
C12:0	0.19	0.23	0.35	0.36	0.05	0.551	0.624
C14:0	1.71	1.74	3.15	2.95	0.35	0.006	0.422
C15:0	0.31	0.35	0.21	0.23	0.03	0.412	0.775
C16:0	20.25	20.41	22.75	22.61	0.54	0.018	0.722
C17:0	0.92	0.91	0.99	0.97	0.05	0.851	0.766
C18:0	18.87	18.76	17.75	17.87	0.68	0.755	0.688
C20:0	0.31	0.33	0.59	0.55	0.07	0.433	0.762
C14:1	0.18	0.17	0.15	0.16	0.05	0.358	0.465
C15:1	0.18	0.19	0.21	0.22	0.06	0.588	0.608
C16:1	1.25	1.19	0.98	0.91	0.08	0.041	0.483
C18:1	32.31	32.45	31.48	31.66	0.58	0.444	0.843
C18:1t10	1.12	1.14	1.22	1.24	0.05	0.755	0.442
C18:1t11	0.95	0.97	0.66	0.63	0.07	0.028	0.343
C20:1	0.22	0.21	0.18	0.19	0.05	0.954	0.555
C20:2	0.29	0.28	0.23	0.24	0.04	0.855	0.321
C18:2n-6 cis	10.84	10.78	11.11	11.26	0.66	0.688	0.781
C18:2c9,t11	0.52	0.49	0.25	0.24	0.07	0.022	0.483
C20:3n-6	0.72	0.68	0.48	0.45	0.07	0.588	0.921
C20:4n-6	3.64	3.58	4.02	3.95	0.18	0.672	0.883
C22:4n-6	0.28	0.32	0.21	0.19	0.03	0.544	0.495
C18:3n-3	1.88	1.85	0.85	0.81	0.15	0.005	0.684
C20:5n-3 EPA	0.81	0.78	0.36	0.34	0.05	0.005	0.321
C22:5n-3 DPA	1.26	1.22	0.89	0.84	0.11	0.005	0.455
C22:6n-3 DHA	1.22	1.18	0.86	0.81	0.12	0.022	0.666
SFAs	42.75	42.98	46.04	45.96	0.85	<0.001	0.711
MUFAs	36.31	36.44	34.95	35.15	0.41	0.019	0.355
PUFA n-3	5.17	5.03	2.96	2.8	0.38	<0.001	0.211
PUFA n-6	15.77	15.64	16.05	16.09	0.45	0.433	0.475
CLA	0.66	0.64	0.32	0.31	0.11	0.015	0.881
PUFA	20.94	20.67	19.01	18.89	0.53	<0.001	0.395
Δ5-plusΔ6-desaturase	38.65	38.28	36.05	35.5	0.48	0.026	0.431

**Table 4 animals-15-00166-t004:** Effects of the forage-to-concentrate ratio (HF:C = high forage-to-concentrate ratio (65:35); LF:C = low forage-to-concentrate ratio (45:55)) and aging time (11 vs. 18 d) on nutritional indices of *Longissimus thoraci* muscle from Podolian young bulls (mean ± SEM).

	Forage:Concentrate						
	HF:C (*n* = 10)		LF:C (*n* = 10)			*p*, Effects	
Aging Time (d)	11	18	11	18	SEM	Diet	Aging
Qualitative indices							
P/S	0.49	0.48	0.41	0.41	0.02	0.033	0.338
n6/n3	3.05	3.11	5.42	5.75	0.25	<0.001	0.466
EPA + DHA (%)	2.03	1.96	1.22	1.15	0.18	0.004	0.511
Unsaturation index	97.55	96.6	88	87.24	0.35	0.041	0.285
Nutritional indices							
Atherogenic index	0.48	0.48	0.66	0.64	0.04	0.023	0.611
Thrombogenic index	0.97	0.99	1.26	1.27	0.04	0.017	0.387
Hypocholesterolemic/hypercholesterolemic	2.63	2.59	2.17	2.15	0.08	0.022	0.421

Unsaturation index = (% monoenoic) + (2 × % dienoic) + (3 × % trienoic) + (4 × % tetraenoic) + (5 × % pentaenoic) + (6 × % hesaenoic). Atherogenic index = [C12:0 + (4 × C14:0) + C16:0]/(ΣUFA). Thrombogenic index = (C14:0 + C16:0 + C18:0)/[(0.5 × MUFA) + (0.5 × Σn-6 PUFA) + (3 × Σn-3 PUFA) + (Σn-3 PUFA/Σn-6 PUFA)]. Hypocholesterolemic/hypercholesterolemic = (C18:1 + ΣPUFA)/(C12:0 + C14:0 + C16:0).

**Table 5 animals-15-00166-t005:** Effects of the forage-to-concentrate ratio (HF:C = high forage-to-concentrate ratio (65:35); LF:C = low forage-to-concentrate ratio (45:55)) and aging time (11 vs. 18 d) on the bioactive compounds of *Longissimus thoracis* muscle from Podolian young bulls (mean ± SEM).

	Forage:Concentrate						
	HF:C (*n* = 10)		LF:C (*n* = 10)			*p*, Effects	
Aging Time (d)	11	18	11	18	SEM	Diet	Aging
Creatine (mg/g)	3.15	3.28	2.83	2.88	0.11	<0.001	0.421
Carnosine (mg/g)	3.68	3.77	2.31	2.45	0.14	0.003	0.385
Anserine (mg/g)	1.25	1.23	1.11	1.09	0.04	0.021	0.457
Creatinine (mg/g)	0.44	0.39	0.36	0.39	0.03	0.444	0.657

**Table 6 animals-15-00166-t006:** Effects of the forage-to-concentrate ratio (HF:C = high forage-to-concentrate ratio (65:35); LF:C = low forage-to-concentrate ratio (45:55)) and aging time (11 vs. 18 d) on the oxidative stability of *Longissimus thoracic* muscle from Podolian young bulls (mean ± SEM).

	Forage:Concentrate						
	HF:C (*n* = 10)		LF:C (*n* = 10)			*p*, Effects	
Aging Time (d)	11	18	11	18	SEM	Diet	Aging
MDA (mg/kg)	0.15	0.17	0.14	0.16	0.10	0.288	0.427
Peroxidability index	45.84	44.18	33.77	32.57	0.58	0.008	0.584

Peroxidability index = (% monoenoic × 0.025) + (% dienoic × 1) + (% trienoic × 2) + (% tetraenoic × 4) + (% pentaenoic × 6) + (% hexaenoic × 8).

## Data Availability

Data supporting the findings of this study are available from the corresponding author upon reasonable request.

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
