# Peer review of "Impact of Dietary-Forage-to-Concentrate Ratio on Podolian Young Bulls’ Performance and Nutritional Properties of Meat"

_animals, 2025, doi:10.3390/ani15020166_

Round 1

Reviewer 1 Report

Comments and Suggestions for Authors

There has been a substantial amount of research on the impact of different concentrate-to-forage ratios in feeding on meat quality. The innovation of this paper lies in adjusting the concentrate-to-forage ratio while controlling for energy and protein, thus there is still a necessity for research. I have the following suggestions:

  1. The research content of this paper also involves growth performance and carcass characteristics; therefore, it is recommended to supplement this information in the title at Lines 2-3 to improve the clarity of the title.

  2. In the keywords section at Lines 30-31, please arrange them in alphabetical order, and it is suggested to add "carcass characteristics" to this section.

  3. In the introduction section at Lines 63-69, it is recommended to merge these paragraphs into one. Additionally, this part should also include a discussion on the impact of dietary concentrate-to-forage ratios on meat quality in recent years, which can be found in authoritative journals on meat quality.

  4. In the Materials and Methods section at Lines 81-83, the authors should explain their reasons for choosing these two specific concentrate-to-forage ratios, rather than ratios that are higher, lower, or in between.

  5. At Line 121, the HP-88 chromatographic column of 100 meters should separate more fatty acids; of course, this is greatly related to the standards used by the authors. Additionally, the authors should specify whether the internal standard used is C17:0 or C19:0, as this is crucial for the accuracy of the results.

  6. At Line 174 in the results section, it appears that this part includes discussion. It should be written as "Results and Discussion." Please ask the authors to verify and modify accordingly.

  7. Table 2 should include the indicator of feed conversion efficiency.

  8. The annotations in Table 4 should use full text instead of the current abbreviations to avoid confusing readers.

  9. In Table 6, the interaction effects of Diet and Aging can also be analyzed.

Author Response

Reviewer 1

Thank you very much for taking the time to review this manuscript. Please find the detailed responses below and the corresponding revisions  highlighted in red in the re-submitted files
There has been a substantial amount of research on the impact of different concentrate-to-forage ratios in feeding on meat quality. The innovation of this paper lies in adjusting the concentrate-to-forage ratio while controlling for energy and protein, thus there is still a necessity for research. I have the following suggestions:
1.    The research content of this paper also involves growth performance and carcass characteristics; therefore, it is recommended to supplement this information in the title at Lines 2-3 to improve the clarity of the title.
AU: Title has been modified accordingly 

2.    In the keywords section at Lines 30-31, please arrange them in alphabetical order, and it is suggested to add "carcass characteristics" to this section.
AU: Keywords have been arranged in alphabetical order

3.    In the introduction section at Lines 63-69, it is recommended to merge these paragraphs into one. Additionally, this part should also include a discussion on the impact of dietary concentrate-to-forage ratios on meat quality in recent years, which can be found in authoritative journals on meat quality.
AU: The two paragraphs has been merged, discussion on the impact of dietary concentrate-to-forage ratios on meat quality has been added (see pag. 2, lines 66-67).

4.    In the Materials and Methods section at Lines 81-83, the authors should explain their reasons for choosing these two specific concentrate-to-forage ratios, rather than ratios that are higher, lower, or in between.
AU. The levels of two forage to concentrate ratios were chosen according to our previous study (Marino et al. 2006). The aim of this choice was to improve the sustainability of beef production systems by replacement of protein concentrates, which are expensive, with locally produced forages. The reason has been added in the text (see pag. 2, lines 87-88).

5.    At Line 121, the HP-88 chromatographic column of 100 meters should separate more fatty acids; of course, this is greatly related to the standards used by the authors. Additionally, the authors should specify whether the internal standard used is C17:0 or C19:0, as this is crucial for the accuracy of the results.             
AU: According to O’Fallon‘s method (reference 20 in the revised manuscript) C13:0 as internal standard was used.

6.    At Line 174 in the results section, it appears that this part includes discussion. It should be written as "Results and Discussion." Please ask the authors to verify and modify accordingly.
AU: Thank you for pointing this out, the title of section 3 has been modified.

7.    Table 2 should include the indicator of feed conversion efficiency.
AU: Feed conversion efficiency has been added

8.    The annotations in Table 4 should use full text instead of the current abbreviations to avoid confusing readers. 
AU: Full text has been added in the annotations of table 4

9.    In Table 6, the interaction effects of Diet and Aging can also be analyze
AU: Interaction effect of diet x aging was not significant for this reason has not been added in the table. 

Reviewer 2 Report

Comments and Suggestions for Authors

The whole paper need a revision in formatation and spaces between words checked

Title: based on the paper, the used term "different" do not represent so well the manuscript. Once the different brings the ideia that will be more than two proportions.

L13: How much longer? Could have this information, a parallel between the common age and the the longer one

L16-17: Which properties?

L19: I don't know if this proportion could be considered low. Compared to what? In feedlot?

L27: There is a comma missing.

L27: Supplementation? What supplementation meaning? The diet is based on concentrate? And the lacking nutrients are supplemented with forage? This contradict with what you stated an introduction (L36-37 and L44-47). 

L26: The increase in creatinine is a good thing? Interesting

Keywords: Must be in alphabetical order

Introduction: I do not think that your introduction explained well the aim of your article. Should have an explanation of the effect of the forage to concentrate proportions, affecting the ruminal microbiotic, fermentation compounds synthesis and their impact on muscule synthesis/production. Also, you did not explained the reaseon of this cattle breed last longer to be finished.

L38: You should define what higher proportion of concentrate meaning, once there is a lot of production systems worldwide

L44: PUFA must be defined in the first use

L53-55: Once the creatinine is formed due to the metabolization of creatine, in a energetic metabolism, if there is an increase in creatinine, as stated in abstract, how this can be good? This wouldn't meaning a higher increase in creatine degratation, what would mean a greater muscle metabolism? Did you correltated this metabolism to the 3-metil hystidin concentration? To base you information that the increase is due to a increase in growt and not to the proteic turnover or muscle degratation due a net energy deficience?

L90: Growth requirement, this need to be more especific. What was the average daily gain aimed? For how long (142 days based on what? The aimed body weight for slaughter or to last this time?)

Title of tabels: Title of tables do not have final point. Also, edit your manuscript to let the title out of the table format.

Table 1: Concentrate, what is their proportions?

L103: 130 days, wouldn't be 142 days? This need explatation and more details.

Item 3. Results. This is result and discussion, not only result

Table 2: Interesting results

L184-188: This citation do not help you to explain your result. This couldn't be due to the fact that the two diets were similar in the requeriments meeting criteria?

L196: So, this is low forage offer? Based on this, your low forrage diet was a little high, don't you think?

L233: Crude fiber? In ruminants? Using Van Soest as source of analysis methodology? Realy?

Table 3: Should must be prior to the results descrition and discussion than in the formatation it is

Table 4 (which you didn't call at the text): Caption shouldn't be in the table, should be out of the table

Table 5: Why didn't you showed the numeric values of the significant variables?

In abstract you stated that creatinine was significant.

L308: The symble seems to be wrong

Results and DISCUSSION: I felt a huge lack of explanations about the changes in forage to concentrate ratio alterations on rumen metabolism, tissue metabolism. There is the result description, a work or two corroborating with result, but not a proper explanation. There is points were the explanation seems to ocorrer, but in the format that is put, got lost with others results description and "discussion" 

L343: Appears? After all this results, you can do a better conclusion

Conclusion: Need to be more direct and clear. Not a discussion continuation. Also, the fact that you didn't stated what means longer time of aging, based on the huge diversity of beef production systems, we do not know what is a longer or not longer aging time.

Author Response

Thank you very much for taking the time to review this manuscript. Please find the detailed responses below and the corresponding revisions  highlighted in red in the re-submitted files

The whole paper need a revision in formatation and spaces between words checked. Title: based on the paper, the used term "different" do not represent so well the manuscript. Once the different brings the ideia that will be more than two proportions.
AU: An editing of whole paper has been done; title has been modified accordingly 

L13: How much longer? Could have this information, a parallel between the common age and the the longer one
AU: The aging time has been specify (see pag.1 line 13)

L16-17: Which properties?
AU: Nutritional, it is reported in the text 

L19: I don't know if this proportion could be considered low. Compared to what? In feedlot?
AU: Low forage concentrate group (LF:C) is defined low compared to the other thesis (HF:C). 

L27: There is a comma missing.
AU: Comma has been added

L27: Supplementation? What supplementation meaning? The diet is based on concentrate? And the lacking nutrients are supplemented with forage? This contradict with what you stated an introduction (L36-37 and L44-47). 
AU: Thank you for pointing this out, supplementation has been replaced with diet 

L26: The increase in creatinine is a good thing? Interesting
AU: As reported in table 6 we found an increase in creatine not in creatinine. The sentence has been modified 

Keywords: Must be in alphabetical order
AU: Keywords have been arranged in alphabetical order

Introduction: I do not think that your introduction explained well the aim of your article. Should have an explanation of the effect of the forage to concentrate proportions, affecting the ruminal microbiotic, fermentation compounds synthesis and their impact on muscule synthesis/production. Also, you did not explained the reason of this cattle breed last longer to be finished.
AU: Introduction has been improved (see pag.1 lines 39-42 and pag. 2 lines 66-67).
A finishing period of 142 days  is  usually used for rustic breed as Podolian cattle in order to reach an adequate body weight for slaughter.

L38: You should define what higher proportion of concentrate meaning, once there is a lot of production systems worldwide
AU: This sentence has been deleted. The proportion of concentrate has been defined with a new sentence (see lines 39-40).

L44: PUFA must be defined in the first use
AU: PUFA has been defined in the text (pag.2 line 47).

L53-55: Once the creatinine is formed due to the metabolization of creatine, in a energetic metabolism, if there is an increase in creatinine, as stated in abstract, how this can be good? This wouldn't meaning a higher increase in creatine degratation, what would mean a greater muscle metabolism? Did you correltated this metabolism to the 3-metil hystidin concentration? To base you information that the increase is due to a increase in growt and not to the proteic turnover or muscle degratation due a net energy deficience?
AU:I agree with reviewer. We didn’t find an increase of creatinine, there was a mistake in the abstract section we wrote creatinine instead of creatine.

L90: Growth requirement, this need to be more specific. What was the average daily gain aimed? For how long (142 days based on what? The aimed body weight for slaughter or to last this time?)
AU: This statement refers to the amount of forage and concentrate that was administrated periodically and increased in each group in order  to satisfy the growth requirement suggested by I.N.R.A. and without changing the ingredients and the forage  to concentrate ratio.
ADG was calculated to evaluate the growth performance during all the experimental trial that lasted 142 days. A finishing period of 142 days  is  usually used for rustic breed as Podolian cattle in order to reach an adequate body weight for slaughter.

Title of tables: Title of tables do not have final point. Also, edit your manuscript to let the title out of the table format.
AU: We have, accordingly modified all the tables. 

Table 1: Concentrate, what is their proportions?
AU: The proportions of ingredients of concentrate has been reported in the footnote.

L103: 130 days, wouldn't be 142 days? This need explatation and more details.
AU: Thank you for pointing this out, the experimental trail lasted 142 days.

Item 3. Results. This is result and discussion, not only result
AU: Thank you for pointing this out; discussion has been added in the section’s title

Table 2: Interesting results
AU: Thank you

L184-188: This citation do not help you to explain your result. This couldn't be due to the fact that the two diets were similar in the requeriments meeting criteria
AU: The  result was explained at lines 180-184,  this citation was added to strengthen our results. However discussion has been improved (see pag. 5, lines 185-189) 

L196: So, this is low forage offer? Based on this, your low forage diet was a little high, don't you think?
AU: In this study the levels of two forage to concentrate ratios were chosen according to our previous study (Marino et al. 2006). The aim of this choice was to improve the sustainability of beef production systems by replacement of protein concentrates, which are expensive, with locally produced forages. I know that forage percentage of LF:C group could be high but it was lower than HF:C.

L233: Crude fiber? In ruminants? Using Van Soest as source of analysis methodology? Realy?
AU: We didn't determine crude fiber but NDF and ADF according to Van Soest.  “Crude fiber” has been modified, accordingly.

Table 3: Should must be prior to the results descrition and discussion than in the formatation it is
AU: Table 3 has been inserted prior to the results description and discussion.

Table 4 (which you didn't call at the text): Caption shouldn't be in the table, should be out of the table 
AU: The reference to table 4 was in the text (line 262 of original manuscript). Caption has been out of the table.

Table 5: Why didn't you showed the numeric values of the significant variables?
AU: p-values have been added.

In abstract you stated that creatinine was significant.
AU: Creatinine was not significant, the creatine was significant. As reported above, in the abstract section we wrote creatinine instead of creatine, the mistake has been corrected in the abstract.

L308: The symble seems to be wrong
AU: symbol has been corrected.

Results and DISCUSSION: I felt a huge lack of explanations about the changes in forage to concentrate ratio alterations on rumen metabolism, tissue metabolism. There is the result description, a work or two corroborating with result, but not a proper explanation. There is points were the explanation seems to ocorrer, but in the format that is put, got lost with others results description and "discussion" 
AU: Discussion on tissue metabolism and rumen metabolism has been improved (see lines 185-199, 194-196 and 239-248).

L343: Appears? After all this results, you can do a better conclusion 
AU: Thank you for pointing this out, the sentence has been modified.

Conclusion: Need to be more direct and clear. Not a discussion continuation. Also, the fact that you didn't stated what means longer time of aging, based on the huge diversity of beef production systems, we do not know what is a longer or not longer aging time. 
AU: Conclusion has been improved (see lines 356-361). Information on aging time have been added (line 363).

Reviewer 3 Report

Comments and Suggestions for Authors

The purpose of this research is to evaluate the effects of varying forage:concentrate on meat quality of beef from Podolian bulls. Overall, the research is interesting and was conducted in a logical fashion. However, the paper could benefit from a more thorough proof reading job and editing (typos and improper grammar). Line by line comments in regards to editing are provided in the comments on the quality of English language section.

Other comments:

-Consider adding bull to the title for readers unfamiliar with Podolian meat 

-Section 2.1.: Were the individual animal pens covered? If so please state. If not, please provide information if on days it rained data data was not included.

-Section 2.2.: Authors mention that FE and DMI were recorded, but that data is not provided anywhere in the manuscript. I would suggest providing that to strengthen the paper. 

-Section 2.2.: How long after slaughter were LT samples collected?

-Tables: Please provide exact p-values. Additionally, tables should be able to stand on their own. In the table footnotes please provide information relevant to study (treatments (65% not just High Forage, Animal Number, etc.) 

Comments on the Quality of English Language

Overall, the paper could benefit from a more thorough in regards to proof reading and editing (typos and improper grammar).

Line by line comments:

Line 9: "Improve" should be improving

Line 12: "great oxidative" should be greater oxidative

Line 16: "affecting also nutritional" should be also affecting the nutritional 

Line 34: Comma after systems

Line 38: "the higher stocking" should be higher stocking. I would suggest revising this entire sentence as it reads weird.

Line 40: is should be are

Line 45: expecially should be especially

Line 71: Delete the second period

I quit correcting grammar after line 45. I would suggest thoroughly proof-reading this manuscript prior to resubmission. 

Line 38: 

Author Response

Thank you very much for taking the time to review this manuscript. Please find the detailed responses below and the corresponding revisions  highlighted in red in the re-submitted files

Reviewer 3
The purpose of this research is to evaluate the effects of varying forage: concentrate on meat quality of beef from Podolian bulls. Overall, the research is interesting and was conducted in a logical fashion. However, the paper could benefit from a more thorough proof reading job and editing (typos and improper grammar). Line by line comments in regards to editing are provided in the comments on the quality of English language section.
Other comments:
-Consider adding bull to the title for readers unfamiliar with Podolian meat 
AU: Young bulls has been added in the title. 

-Section 2.1.: Were the individual animal pens covered? If so please state. If not, please provide information if on days it rained data data was not included.
AU: The pens were covered, this information has been added in the text (see line 93).

-Section 2.2.: Authors mention that FE and DMI were recorded, but that data is not provided anywhere in the manuscript. I would suggest providing that to strengthen the paper. 
AU: FE and DMI were added in the revised manuscript 

-Section 2.2.: How long after slaughter were LT samples collected?
AU:LT was collected after 24 hours and then vacuum aged for 11 and 18 days 

-Tables: Please provide exact p-values. Additionally, tables should be able to stand on their own. In the table footnotes please provide information relevant to study (treatments (65% not just High Forage, Animal Number, etc.)
AU: p-values has been added. All tables have been modified accordingly

Comments on the Quality of English Language
Overall, the paper could benefit from a more thorough in regards to proof reading and editing (typos and improper grammar).
Line by line comments:
Line 9: "Improve" should be improving
AU: Done

Line 12: "great oxidative" should be greater oxidative
AU: Done

Line 16: "affecting also nutritional" should be also affecting the nutritional 
AU: Done

Line 34: Comma after systems
AU: Done

Line 38: "the higher stocking" should be higher stocking. I would suggest revising this entire sentence as it reads weird.
AU: Done

Line 40: is should be are
AU: Done

Line 45: expecially should be especially 
AU: Done

Line 71: Delete the second period 
AU: Done

I quit correcting grammar after line 45. I would suggest thoroughly proof-reading this manuscript prior to resubmission. 

Round 2

Reviewer 2 Report

Comments and Suggestions for Authors

Ok

Author Response

The authors have made some corrections and improvements to the paper's quality. However, I still find the introduction lacking in properly defining what constitutes a high- and low-forage diet.
For instance, globally, a diet with 65% concentrate could be considered more concentrate than forage, but many production systems use even higher concentrate levels.
Additionally, the authors did not adequately explain why the animals had an extended finishing period. They could attribute this to factors like rusticity or growth metabolism.

AU: Thank you very much for taking the time to review this manuscript. 
As reported in the revised paper the levels of two forage to concentrate ratios were chosen according to our previous study in which two different forage to concentrate ratio were compared (60:40 vs 70:30).
In this paper, the aim of the choice 65:35 vs 45:55 of forage to concentrate ratio was to improve the sustainability of beef production systems by replacement of protein concentrates, which are expensive, with locally produced forages as alfalfa hay. According to the reviewer, a diet  with 65% concentrate, can be considered a low concentrate diet in breed different from Podolian and according to different production systems.However, if we considered Podolian, as in the present experiment, and the Podolian production systems, a diet with 55% concentrate can be considered a low forage diet. In addition, Podolian breed is a rustic breed, as reported in introduction, so a finishing period of 142 days is usually used for this rustic breed in order to reach an adequate body weight for slaughter. 
These points have been explained in the introduction of the revised paper R2 (see lines 41-43, 53-55,78-81).
